# STRUCTURED FINE-TUNING ENABLES DATA-EFFICIENT ADAPTATION OF CODE LANGUAGE MODELS

## ABSTRACT

Current models tailored for code tasks often adopt the successful pre-training-then-fine-tuning paradigm from natural language processing, treating source code in plain text as in natural language. This approach, however, overlooks the well-defined and unambiguous structures inherent in programming languages. In this work, we explore a data-efficient adaptation of pre-trained code language models by further training and fine-tuning them with program structures, which significantly improve the performance of the downstream coding tasks. Specifically, we represent programs as parse trees, also known as concrete syntax trees (CSTs), and refine a model with serialized CSTs. Fine-tuning with structures encourages the model to learn not only the associations of code text in different languages but also the mappings of their structures and grammars, by using only a small amount of data (e.g., 100 examples). With a focus on generation, we design training objectives for encoder-decoder and decoder-only architectures. We rigorously evaluate the proposed approach on various coding tasks and demonstrate that integrating parse structures with the plain-text representation of source code offers notable advantages, particularly in scenarios of low-data code translation.

## 1 INTRODUCTION

Natural language models consider text as a sequence of tokens, which can be encoded and decoded using neural sequence models. With the attention mechanism considered to be "all you need" (Vaswani et al., 2017), natural language models are extended to programming languages by treating source code as plain text, such that one would conveniently reuse existing self-learning approaches (e.g., masked-token recovery, next-token prediction) for pretraining and fine-tuning. Adapting models from natural languages to code requires mainly curating training data that includes sufficient code. This simple adaptation works well, as demonstrated by the remarkable performance of various code language models (Wang et al., 2021; Chen et al., 2021; Li et al., 2023).

We argue, however, that source code inherits well-defined structures that may be leveraged to enhance the performance of coding tasks. These structures unveil the compositional semantics of code, thereby fostering the learning of compositional solutions within neural models, if properly modelled. Unlike natural languages, where obtaining syntax structure can be challenging due to intrinsic ambiguity, programming languages come with grammar and compilers that can unambiguously and efficiently parse a program. The resulting parse tree, a concrete syntax tree (CST), is only one example of a program's various *structural representations*. Other well-studied structures in the field of program analysis include, for example, abstract syntax trees (ASTs), control flow graphs (Allen, 1970), program dependence graphs (Ferrante et al., 1987), code property graphs (Yamaguchi et al., 2014), control dependence graphs (Cytron et al., 1991), and system dependence graphs (Graf, 2010). All these representations structure a program and offer additional insights beyond the sequence representation. Many of these representations and variants have been exploited for program understanding and vulnerability analysis, often paired with graph neural networks or graph transformers because they are capable of encoding structured data (Allamanis et al., 2018; Zügner et al., 2021). For a similar reason, structural representations should also help code–code translation and text–code generation tasks, when a model learns not only the association of text but also the mapping between structures.

This work advocates using CSTs to adapt pre-trained models for coding tasks. The focus on using CSTs rather than other structures comes with several reasons. First, most important tasks require decoding a program (such as code translation and generation). It is unclear how one can deduce a program from various graph representations of code; it is also challenging if one works on the ASTs. Second, off-the-shelve CST parser generators exist for many languages, and the support for additional languages is expected to grow (Tree-sitter). Tooling support contributes to the feasibility and practicality of using CSTs. Third, being a tree, a CST can be converted to and from a sequence by using an easily defined invertible mapping. Such a property allows reusing the model architecture (i.e., Transformer) that encodes and decodes sequences without modification. In this work, we will explore both conversion directions of tree serialization and de-serialization for sequence modeling of code (in Section 3.2).

Building on the notable benefits of CSTs as a structural representation, we advocate their use for continual pre-training and fine-tuning a pre-trained language model for coding tasks. First, we formulate pre-training objectives that leverage the extensive structural information of CSTs, enhancing the structural understanding of code within pre-trained sequence models. Subsequently, we develop structured fine-tuning that enables quick adaptation at minimal costs, requiring significantly less data than plain-text code. This approach is particularly advantageous when working with low-resource languages where monolingual data is sparse and parallel data is even more scarce; it is also beneficial when collecting sufficient data for a downstream task is difficult. While in-context learning offers an alternate solution, it often falls short, especially when accessing a large model is beyond one's limited budget.

In this work, we demonstrate that in the low-data scenario, fine-tuning with structures can outperform not using structures by as much as 15 BLEU points and 8 CodeBLEU points for the Java-C# code translation task (both directions) using only 100 training examples. Similarly, in the text-to-code generation task for the MBPP dataset, employing structures with only 100 training examples significantly improves the pass@1 metric, increasing it from 1.68% to 4.40%. The contributions of this work can be summarized as follows.

1. We introduce a novel approach for the structured pre-training and fine-tuning of pre-trained language models. In particular, we explore the serialized form of CST, considering its suitability as an output format of code for decoding, beyond merely facilitating representation learning.

2. Our structured pre-training and fine-tuning method is applicable to both encoder-decoder and decoder-only models. Through empirical experiments with both types of models, we have observed consistent and beneficial results from our approach.

3. We demonstrate that structured fine-tuning allows data-efficient adaptation of pre-trained models, compared with a usual fine-tuning that treats source code as plain text. In particular, with small training data, using structures generally improves the performance of a variety of tasks, including code translation, generation, and summarization.

## 2 Related Works

**Machine Learning for Code**   Early approaches applied statistical learning methods to programming tasks (Nguyen et al., 2013; Movshovitz-Attias & Cohen, 2013; Raychev et al., 2014; Allamanis et al., 2014). With the advent of deep learning, deep neural networks gradually replaced pure statistical approaches (Allamanis et al., 2016; Mou et al., 2016; Gu et al., 2016; Iyer et al., 2016). However, these models were often task-specific and were trained on particular datasets, limiting their adaptability to custom tasks. More recently, with the increasing popularity of pre-trained models in natural language processing, similar pre-trained models for code were introduced. Among the pioneering efforts, Kanade et al. (2020) trained a BERT-based model on a large-scale Python dataset and showed improved performance on multiple downstream tasks with limited fine-tuning. Subsequently, multiple pre-trained models with varying architectures, data composition, and code representations were proposed. Notable examples include CodeBERT (Feng et al., 2020), GraphCodeBERT (Guo et al., 2020), PLBART (Ahmad et al., 2021), CodeT5 (Wang et al., 2021), UniXCoder (Guo et al., 2022), CodeGen (Nijkamp et al., 2022), and StarCoder (Li et al., 2023).

**Incorporating Structured Representations for Code**   Given that code can be represented by structures such as syntax or parse trees, data flows, and execution flows, several works explored

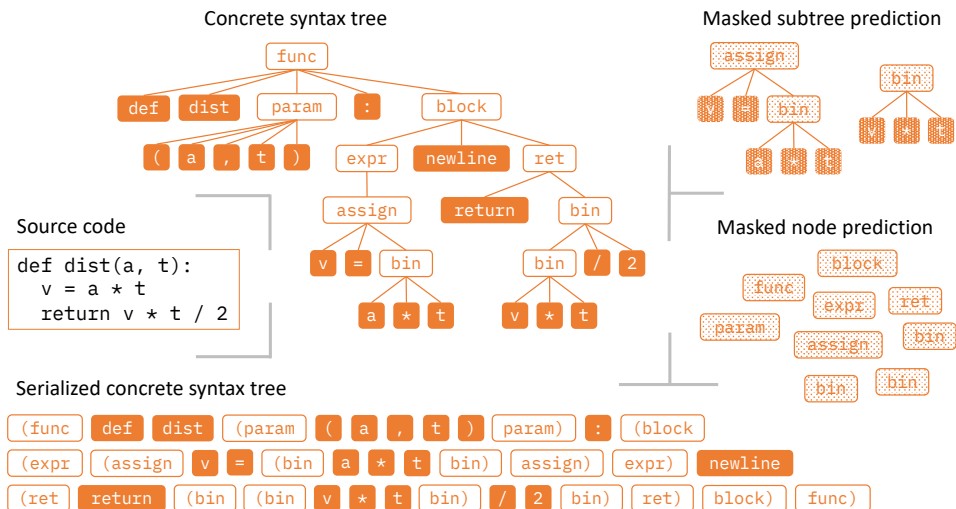

Figure 1: A representative Python program along with its CST (simplified for illustration) in the tree and serialized forms, respectively. Also illustrated are the masked subtree prediction and masked node prediction training objectives we propose to adapt encoder-decoder models to code structures.

various approaches to integrate these structures into code models. For instance, Mou et al. (2016) introduced a convolutional kernel applied to the program's AST. Phan et al. (2017) proposed leveraging the program's execution flow in combination with graph-based CNNs for software defect prediction. Meanwhile, Hu et al. (2018) proposed converting an AST into a sequence using tree traversal and employing LSTM to generate code comments. Zhou et al. (2019) utilized composite code representations including the AST, control flow, data flow, and natural code sequence for vulnerability detection. To incorporate code structures into the Transformer architecture, prior works introduced novel positional encodings that represent node positions within the tree (Shiv & Quirk, 2019; Peng et al., 2022); or they utilized syntax or parse tree traversals or path information to provide structural information to the model (Kim et al., 2021; Wang & Li, 2021; Peng et al., 2021; Jiang et al., 2021; Guo et al., 2020). Additionally, specific tree-based attention mechanisms have been proposed (Tang et al., 2022; Wang et al., 2023).

In all the aforementioned works, regardless of whether the code structure is incorporated or not, models treat source code as plain text during downstream tasks. In contrast, we propose to not only encode but also decode serialized CSTs to inject structural information into the model.

## 3 STRUCTURED CODE REPRESENTATION

A compiler transforms the high-level source code written by programmers into low-level machine code that can be executed on a computer's hardware. Broadly speaking, the compilation consists of two phases: the front end (analysis) and the back end (synthesis) (Aho et al., 2007). The front end includes lexical analysis (tokenization), syntax analysis (parsing), and semantic analysis, generating an AST or a similar intermediate representation that captures the program's logical structure and semantics. Therein, grammar specifies how keywords, identifiers, and literals should be structured, which the front-end uses to break the source code into individual tokens and create the syntax tree, ensuring that the source code adheres to the language's syntax rules.

### 3.1 CONCRETE SYNTAX TREE

In more details, compiler front ends typically start by creating a CST from the input source code and then abstract away the non-essential details from the CST to create the AST. A CST is a tree representation of the source code according to the language's grammar. It closely mirrors the code's textual structure and includes all the syntactic details like parentheses and punctuation. In contrast, an AST simplifies the CST, retaining only the program's logical structure.

While ASTs are more commonly used due to the balance they strike between compactness and preserving essential program structures for further analysis and transformations, we use CSTs for program representations for several reasons. First, CSTs faithfully retain the exact syntactic structure, including all punctuations, whitespaces, and formatting. This preservation of syntax is essential when one must provide all the details of the code to the model and precisely reconstruct the code from the model generated CST. Second, CSTs are more generally applicable than ASTs, as they can be built directly from the language's grammar without resorting to language-specific optimizations or implementations required by the ASTs. We do note however that CSTs are typically more verbose than ASTs due to their inclusion of all syntactic details.

## 3.2 Serialization and Deserialization

CST is an n-ary tree, where each node can have a finite but arbitrary number of children. We call the leaf nodes, *terminal nodes*, and the rest, *non-terminal nodes*. The terminal nodes contain all textual information of the code while the non-terminal nodes correspond to grammar rules. In Figure 1, terminal nodes have a filled background and one sees that these nodes are sufficient to deduce the program from the tree. However, one must serialize the tree to make it consumable by a typical Transformer, which in turn admits a guaranteed process for deserialization in subsequent uses.

To this end, we define the serialization as a pre-post-order tree traversal, where a node is visited before its ordered children and is revisited after all its descendants have been visited. Let `Node` denote the current node and let `subtree` denote the subtree rooted at `Node` and excluding `Node`. Then, the serialization reads (recursively) `(_.Node subtree Node._)`, where `(_.Node` and `Node._)` are separate tokens that represent the first and second visit of `Node` in the traversal. See Figure 1 for an example. Explicitly marking the boundary of the subtree by `(_.Node` and `Node._)` is necessary to ensure unambiguous deserialization. For more examples, see Appendix A.

## 4 Adapting Pre-Trained Models to Code Structures

Existing pre-trained models are generally trained on source code treated as plain text without explicit incorporation of the structural information. We propose to adapt these models to code structures by using serialized parse trees for further pre-training and fine-tuning, whenever possible. The fine-tuning part is straightforward. However, fine-tuning only is not sufficient to learn embeddings for the non-terminal nodes. Hence, in this section, we mainly discuss the pre-training part, specifically pre-training tasks. Note that often training data include not only code but also text (e.g., source code with comments). We develop pre-training tasks that effectively learn the structural elements that can go in cohort with the remaining code and text. In what follows, we use $x$, $y$, $z$ to denote natural language text, program text, and the corresponding serialized parse tree, respectively.

### 4.1 Encoder-Decoder Models

**Masked SubTree Prediction (MSP)**   To learn the tree structure, we propose to randomly mask subtrees before encoding and ask the decoder to predict them (see Figure 1). Specifically, we mask 15% of the nodes in the parse tree, by randomly selecting non-terminal nodes and masking their entire subtrees, until the budget is attained. We replace each of the selected subtree with the same mask token and ask the model to predict these subtrees. After serialization, masking subtrees is effectively equivalent to masking the corresponding span from the sequence of the serialized tree. Hence, this technique can be considered a tree extension of the masked span prediction objective popularly used in natural language models (Raffel et al., 2020; Joshi et al., 2020; Wang et al., 2021). The training objective can be written as

$$\mathcal{L}_{\text{MSP}} = \sum_{t=1}^{k} \log p(z_t^{\text{mask}} | z^{\backslash \text{mask}}, z_{<t}^{\text{mask}}), \tag{1}$$

where $z^{\backslash \text{mask}}$ is the masked input and $z_t^{\text{mask}}$ is the masked sequence to predict with length $k$. Since the subtrees include non-terminal and terminal nodes, this task aids the model in learning the relationship between language grammar (non-terminal nodes) and code semantics (terminal nodes).

**Masked Node Prediction (MNP)**    To enable the model to further learn the language grammar, we propose the masked node prediction objective, where all the non-terminal nodes are masked by a unique sentinel token and they together form, in the original order, a sequence $\boldsymbol{I}$. The decoder is then asked to predict $\boldsymbol{I}$ (see Figure 1). This task is similar to the masked identifier prediction objective of Wang et al. (2021) and the DOBF objective of Lachaux et al. (2021), except that the masked tokens here are not the original program tokens. The training objective can be written as

$$\mathcal{L}_{\text{MNP}} = \sum_{t=1}^{|\boldsymbol{I}|} \log p(\boldsymbol{I}_t|\boldsymbol{z}^{\backslash \boldsymbol{I}}, \boldsymbol{I}_{<t}), \tag{2}$$

where $\boldsymbol{z}^{\backslash \boldsymbol{I}}$ denotes the serialized tree with non-terminal nodes masked. We note that this is a much harder task than the MSP objective, because in MNP the majority of the tree is masked and the model has to learn the language grammar to predict them.

**Text-to-Tree (TeTr) and Tree-to-Text Conversion (TrTe)**    When the data contains natural language $\leftrightarrow$ code pairs (e.g., code with the natural language description), we encode either part and decode the other to align both parts. The training objectives can be written as

$$\mathcal{L}_{\text{TeTr}} = \sum_{t=1}^{|\boldsymbol{z}|} \log p(\boldsymbol{z}_t|\boldsymbol{x}, \boldsymbol{z}_{<t}) \qquad \mathcal{L}_{\text{TrTe}} = \sum_{t=1}^{|\boldsymbol{x}|} \log p(\boldsymbol{x}_t|\boldsymbol{z}, \boldsymbol{x}_{<t}), \tag{3}$$

which are similar to the typical objectives $\log p(\boldsymbol{y}_t|\boldsymbol{x}, \boldsymbol{y}_{<t})$ and $\log p(\boldsymbol{x}_t|\boldsymbol{y}, \boldsymbol{x}_{<t})$ used in Wang et al. (2021), except that code $\boldsymbol{y}$ is replaced by seralized tree $\boldsymbol{z}$. These two tasks match the downstream utilization of the model, where it has to either generate code given natural language description or summarize the given code in natural language.

## 4.2    DECODER-ONLY MODELS

For decoder-only models, we find that it is effective to reuse the causal language modelling objective over the serialized tree $\boldsymbol{z}$:

$$\mathcal{L}_{\text{DEC}} = \sum_{t=1}^{|\boldsymbol{z}|} \log p(\boldsymbol{z}_t|\boldsymbol{z}_{<t}). \tag{4}$$

When there exist natural language $\leftrightarrow$ code pairs, we replace $\boldsymbol{z}$ in the above formula by $\boldsymbol{z}' = [\boldsymbol{x} : \boldsymbol{z}]$; that is, concatenating text and code. Compared with the specialized objectives MSP and MNP for encoder-decoder models, here we require the model to reconstruct all tokens (both terminal and non-terminal nodes) in an autoregressive manner.

## 5    EXPERIMENT SETUP

### 5.1    PRETRAINED MODELS AND TOKENIZERS

To evaluate the proposed method, we use two existing pre-trained models, one encoder-decoder and other decoder-only.

- **CodeT5** (Wang et al., 2021) is an encoder-decoder model trained on the CodeSearchNet dataset (Husain et al., 2019) using denoising seq2seq objectives on code data, along with bimodal conversion objectives betwen natural langauge and code. We use the CodeT5-base (220M) model for experiments.
- **CodeGen** (Nijkamp et al., 2022) is a decoder-only model sequentially trained on the English corpus (CodeGen-NL), followed by a code dataset collected from Google's BigQuery (CodeGen-Multi), and then a Python dataset (CodeGen-Mono). We use the CodeGen-Multi (350M) model for experiments.

In addition to adapting the pre-trained models to structures, we augment the respective tokenizer by including the non-terminal nodes, each treated as one token. This allows the model to learn targeted emebeddings for the non-terminal nodes and also makes the input/output length manageable. In the ablation study (Section 7) we will show that without including the non-terminal node tokens in the tokenizer will substantially compromise the downstream performance.

## 5.2 PRE-TRAINING DATASET

We use the CodeSearchNet dataset augmented partially by the Stack dataset (Kocetkov et al., 2022) to further train the pre-trained models. CodeSearchNet contains nearly 6.5 million data samples extracted from the most popular GitHub projects with permissive licenses. Among them, around 2.3 million samples have comments accompanying the code, while the remaining are code segments only. CodeSearchNet contains six programming languages. However, it misses some languages that are used in our downstream tasks; namely, C and C#. To include these languages, we subsample 1 million samples for each from the Stack dataset, which results in a total of 8.5 million samples across eight progamming languages. More details are given in Appendix B.

## 5.3 CONTINUAL PRE-TRAINING

We further pre-train CodeT5 and CodeGen by using the objectives described in Section 4. We train for one epoch, by using 32 V100-32GB GPUs with an effective batch size of 1024. We provide the list of hyper-parameters in Appendix C. It is important to note that the continual pre-training is rather lightweight. Compared with the reported training time of 12 days by using 16 A100-40GB GPUs for the CodeT5 model and 450,000 steps for the CodeGen model, our continual pre-training takes only 15 hours for CodeT5 and approximately 8,000 steps for CodeGen.

## 5.4 DATA-EFFICIENT FINE-TUNING

Existing benchmarks for evaluating code models typically use datasets containing a few thousand to several hundred thousand data samples. For instance, the Java-to-C# code translation dataset in CodeXGLUE (Lu et al., 2021) contain 10.3k samples, while the text-to-code Concode dataset (Iyer et al., 2018) contains 100k samples. While it is possible to curate data at expensive cost in a one-time effort, in many scenarios dataset creation can be financially burdensome and even practically unattainable. This issue is even more pronounced for low-resource languages or domain-specific languages. Hence, we direct our attention on low-data scenarios by using data-efficient approaches. In this context, our experiment setting is to fine-tune models using only a few hundred samples, with the overarching aim of improving downstream performance on small data.

## 5.5 EVALUATION METRICS

Evaluating generative models for code initially followed procedures similar to evaluating text generation. Evaluation method included comparing generated code sample to reference samples using BLEU score (Papineni et al., 2002) or Exact Match (EM). However, EM fails to consider code variability in achieving the same objective, and BLEU's correlation with semantic code correctness is weak, as highlighted by Tran et al. (2019). To address this, Ren et al. (2020) introduced CodeBLEU, a composite metric combining BLEU, weighted n-gram matching, syntactic AST matching, and semantic data flow matching, demonstrating stronger alignment with human-assessed code quality. Along with fuzzy metrics, recent studies, including Roziere et al. (2020) and Kulal et al. (2019), turned to functional correctness assessment. This approach involves generating multiple code samples and deeming the problem solved if any sample passes all test cases. Chen et al. (2021) further refined this method, addressing its high variance with an unbiased estimator. In our work, we adopt the estimator introduced by Chen et al. (2021) to gauge functional correctness.

## 6 RESULTS

To evaluate the effectiveness of our approach, we fine-tune CodeT5 (+structure) and CodeGen (+structure) on five downstream tasks, including code translation, code generation, and code summarization. For each task, we fine-tune the models with a few hundred data examples but evaluate the test accuracy on the complete test set. We repeat each experiment with five random seeds and report mean and standard deviation.

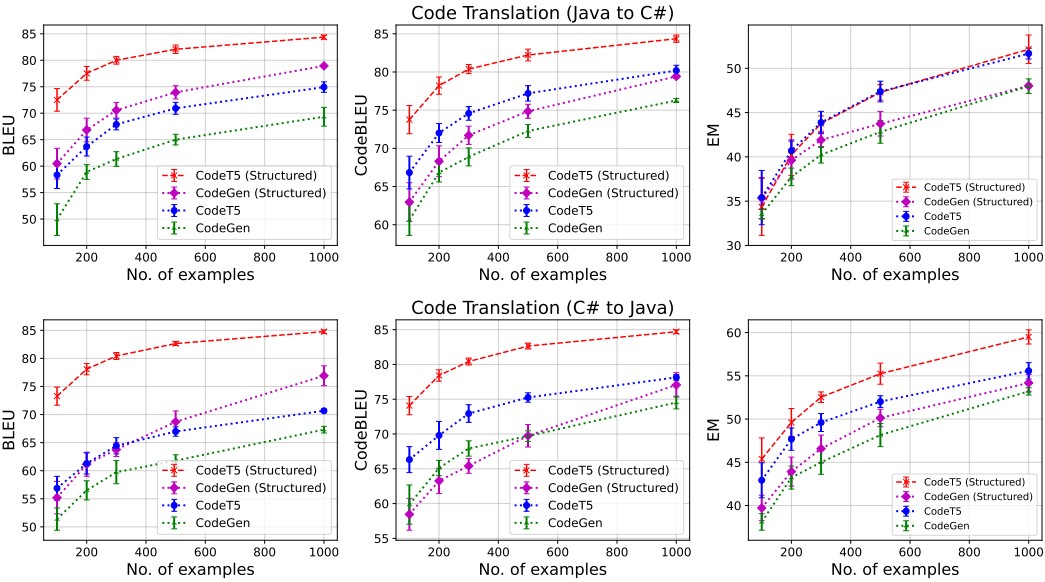

Figure 2: Code translation: Java-to-C# (top) and C#-to-Java (bottom) results for varying number of fine-tuning examples.

## 6.1 CODE TRANSLATION

Code translation benchmarks aim to translate reasonable-length code segments from one programming language to another. Code translation has many practical values, such as in IT modernization, where legacy code needs be rewritten by using a modern language for reducing the maintenance cost. For this task, we use the Java-C# translation dataset available within the CodeXGLUE benchmark (Lu et al., 2021). This dataset contains parallel code between Java and C# on the function level; it was extracted from multiple open-source projects that contain parallel implementations. The results for Java-to-C# and C#-to-Java are shown in Figure 2. We see that both base models perform poorly when fine-tuned with only 100 examples but they keep being improved with more and more examples. Specifically, the encoder-decoder model CodeT5 performs better than the decoder-only model CodeGen. Additionally, we observe that the structured models consistently outperform their base counterparts by a significant margin, in both translation directions. It is worth noting that the greatest improvement appears in CodeT5, where fine-tuning with as few as 100 examples results in an improvement of nearly 15 BLEU score points and 8 CodeBLEU score points in both directions. Similar improvements remain as we progressively increase the number of examples to 1000.

## 6.2 CODE GENERATION

Code generation is the task of generating source code given a natural language description. For this task, we test our method on three datasets: (a) the CoNaLa dataset (Yin et al., 2018), which contains curated pairs of descriptions and Python program snippets mined from StackOverflow; (b) the Concode dataset (Iyer et al., 2018), which consists of paired Java programs and comments collected from GitHub projects; and (c) the Mostly Basic Python Programming (MBPP) dataset (Austin et al., 2021) — a crowd-sourced dataset of Python programs along with descriptions.

In Figure 3, we present the results for these benchmarks. For the CoNaLa and Concode benchmarks, we report the BLEU, CodeBLEU, and the EM scores. On the other hand, for the MBPP benchmark, we report the pass@1 score instead of EM, because this dataset contains unit tests for each data sample.

For the CoNaLa benchmark, we observe that both the CodeT5 (Structured) model and the CodeGen (Structured) model outperform their base variants by nearly 15 BLEU score points and 10 Code-BLEU score points. It is worth highlighting that while the CodeGen base variant underperforms the CodeT5 model on this benchmark, the structured CodeGen counterpart outperforms both the base

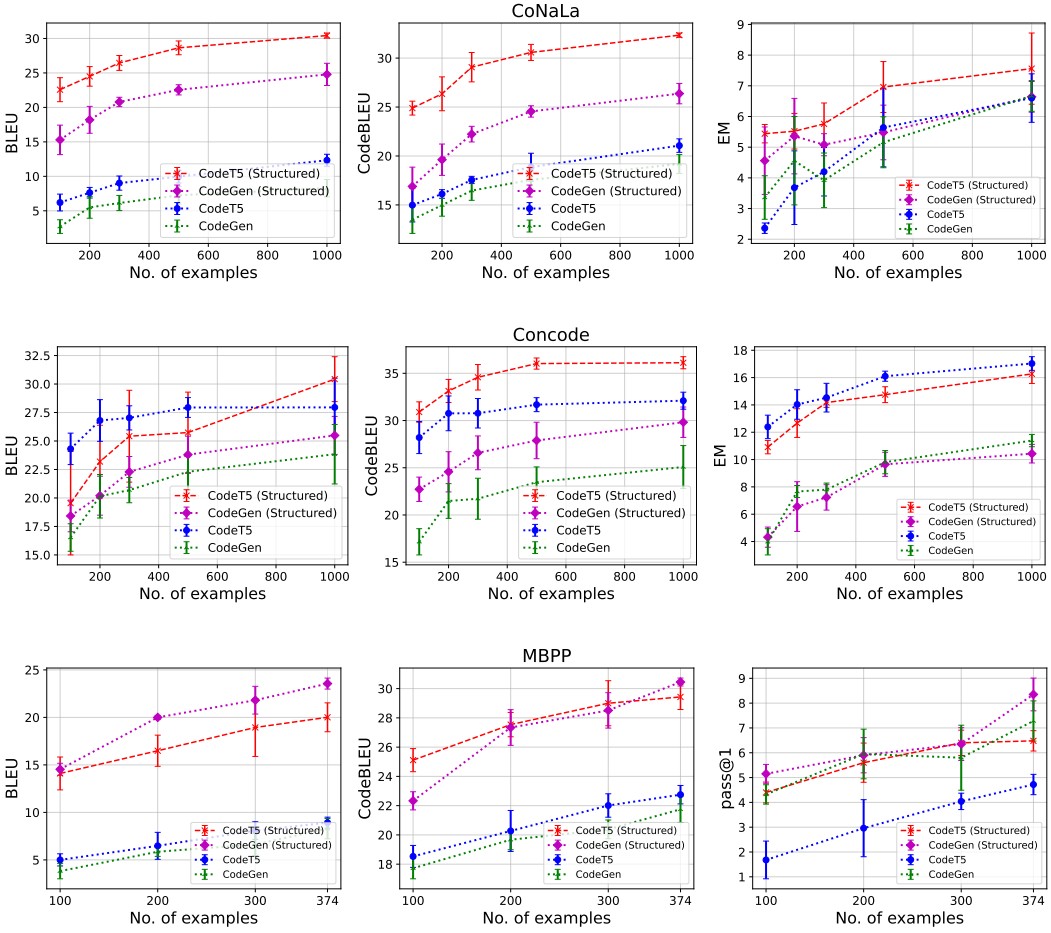

Figure 3: Code generation results for varying number of fine-tuning examples.

variants by a large margin. For the Concode benchmark, we see substantial improvements in the CodeBLEU scores for both structured models, but the results on the BLEU score for the CodeT5 (Structured) model are mixed — lagging behind the base variant for the initial data points. Finally, on the MBPP benchmark, we observe significant improvements for all three metrics. Both models improve the BLEU and CodeBLEU scores, but the CodeT5 (Structured) model also significantly improves the pass@1 metric from 1.68% to 4.40% when utilizing only 100 fine-tuning samples, and improving from 4.72% to 6.48% when utilizing all 374 examples available in the MBPP benchmark. The decoder-only model CodeGen too displays improvements across all three metrics — improvements up to 10 BLEU and CodeBLEU score points, and improving the pass@1 score by 1 percentage point.

## 6.3 CODE SUMMARIZATION

Code summarization is the task of generating a natural language description for a given code segment. We utilize the CodeSearchNet dataset (Husain et al., 2019) to evaluate the performance across six programming languages. In Figure 4, we show the average BLEU scores. Detailed results for each language are provided in Appendix D. We observe that the CodeT5 (Structured) model improves the average performance by 0.5 BLEU score points when fine-tuned with only 100 examples, but this performance gap improves to about 2 BLEU score points when utilizing 500 fine-tuning examples. Similarly, the CodeGen (Structured) model demonstrates improvement over its baseline variant, albeit with relatively modest gains — an improvement of 0.5 BLEU score points over the baseline with 500 fine-tuning examples.

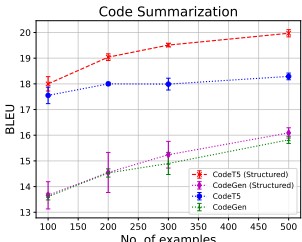

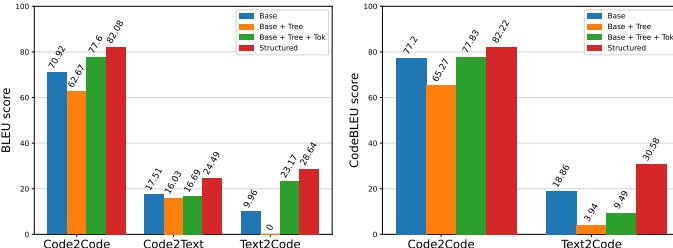

Figure 4: Code summarization fine-tuning results.

Figure 5: Ablation analysis for code translation, code summarization, and code generation tasks.

# 7 ABLATION ANALYSIS

In this section, we aim to better understand the key contributors to the improved performance of structured code models compared to their base variants. We investigate three downstream tasks, namely, code translation from Java to C#, code summarization for Go, and code generation with the Concode dataset. For each of these tasks, we fine-tune the CodeT5 model with 500 examples and present the mean BLEU and CodeBLEU scores over five random seeds in Figure 5. We note the observations below.

**Fine-tuning base models on serialized parse trees only results in performance drop** For this experiment, instead of fine-tuning the base model on source code as text, we fine-tune it on serialized parse trees. This allows us to understand the role of serialized trees alone in the model performance. Interestingly, we observe a significant drop in performance for all three tasks as compared to the base variant, indicating that naively fine-tuning the base models on serialized parse trees does not result in any performance gain. We note that the key reasons behind this performance drop are: (a) the non-terminal nodes being split up into multiple sub-tokens resulting in context loss, and (b) increased length of input/output sequences resulting in greater truncation of context.

**Adding new non-terminal tokens to the base model results in performance gain** We next experiment with adding the non-terminal nodes in the parse tree as special tokens to the tokenizer and then fine-tuning the model. With the updated tokenizer, we allow the model to learn specialized embeddings for non-terminal nodes and also allow the length of the input/output sequences to be manageable. We note that adding non-terminal nodes as special tokens in the tokenizer helps the model achieve better performance. However, as compared to the base model, the results are mixed. On translation tasks, the performance is better than the base model, but for summeriazation and generation tasks the performance lags behind the base model.

**Continual pre-training provides additional performance gains** We next compare the performance of the structured model – model with the updated tokenizer and adapted to structures through continued pre-training – with the variants noted above. We find that the structured model achieves the best performance across all tasks and metrics, suggesting that both the updated tokenizer and the pre-training objectives play important roles in improving the model performance across all tasks.

# 8 CONCLUSIONS

In this work, we explore data-efficient fine-tuning of code language models by utilizing the serialized parse tree of source code. We develop training objectives for both encoder-decoder and decoder-only models to enable the adaptation of existing pre-trained models to code structures. Evaluating the approach on multiple downstream tasks, including code translation, generation, and summarization, we observe significant gains in both fuzzy metrics (BLEU and CodeBLEU scores) and functional metrics (pass@$k$ score), especially in low-data scenarios.

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

## A CONCRETE SYNTAX TREE – SERIALIZATION AND DESERIALIZATION

In this section, we provide two examples of source code along with their corresponding serialized CSTs that is provided as input to the model. The first example is a Python program (Figure 6a and Listing 1), while the second example is a Go program (Figure 6b and Listing 2). We note that we format the serialized tree is a program-like format for easy understanding by the reader. In the actual serialization, there are no indentations and newlines.

```
def add_nums(a, b):
  c = a + b
  return c
```

(a) Python

```
func main() {
  fmt.Println("7.0/3.0 =", 7.0/3.0)
  fmt.Println(true && false)
}
```

(b) Go

Figure 6: Example source code.

```
 1 (_.module
 2     (_.function_definition
 3         def
 4         (_.identifier add_nums identifier._)
 5         (_.parameters
 6             (
 7             (_.identifier a identifier._)
 8             ,
 9             (_.identifier b identifier._)
10             )
11         parameters._)
12         : indent
13         (_.block
14             (_.expression_statement
15                 (_.assignment
16                     (_.identifier c identifier._)
17                     =
18                     (_.binary_operator
19                         (_.identifier a identifier._)
20                         +
21                         (_.identifier b identifier._)
22                     binary_operator._)
23                 assignment._)
24             expression_statement._)
25             newline
26             (_.return_statement
27                 return
28                 (_.identifier c identifier._)
29             return_statement._)
30             newline
31             dedent
32         block._)
33     function_definition._)
34 module._)
```

Listing 1: Python code serialized CST

```
1  (_.function_declaration
2      func
3      (_.identifier main identifier._)
4      (_.parameter_list ( ) parameter_list._)
5      (_.block
6          {
7          (_.call_expression
8              (_.selector_expression
9                  (_.identifier fmt identifier._)
10                 .
11                 (_.field_identifier Println field_identifier._)
12             selector_expression._)
13             (_.argument_list
14                 (
15                 (_.interpreted_string_literal "7.0/3.0_="
   interpreted_string_literal._)
16                 ,
17                 (_.binary_expression
18                     (_.float_literal 7.0 float_literal._)
19                     /
20                     (_.float_literal 3.0 float_literal._)
21                 binary_expression._)
22                 )
23             argument_list._)
24         call_expression._)
25         \n\n
26         (_.call_expression
27             (_.selector_expression
28                 (_.identifier fmt identifier._)
29                 .
30                 (_.field_identifier Println field_identifier._)
31             selector_expression._)
32             (_.argument_list
33                 (
34                 (_.binary_expression true && false binary_expression._)
35                 )
36             argument_list._)
37         call_expression._)
38         \n }
39     block._)
40 function_declaration._)
```

Listing 2: Go code serialized CST

## B  PRE-TRAINING DATASET STATISTICS

We provide the pre-training dataset statistics in Table 1. We utilize 8.5 million data samples to pre-train the model, of which, 2.3 million have natural language along with the code segment. The remaining 6.2 million data samples have only the code segment.

## C  HYPER-PARAMETERS

In Table 2, we provide the list of hyper-parameters used to pre-train code models. In addition to the values listed in the table, we train the models in fp16 precision to speed-up training. For the CodeT5 model we set the input and output sequence lengths to 512 maximum tokens. For CodeGen model, we use a context length of 1024 tokens. In Table 3, we provide the list of hyper-parameters used for data-efficient fine-tuning.

Table 1: Pre-Training data statistics

| Language | with NL | w/o NL | Total |
|---|---|---|---|
| Go | 347,665 | 379,103 | 726,768 |
| Ruby | 53,497 | 110,551 | 164,048 |
| Python | 499,055 | 657,030 | 1,156,085 |
| Java | 499,618 | 1,070,271 | 1,569,889 |
| JavaScipt | 139,902 | 1,717,933 | 1,857,835 |
| PHP | 579,763 | 398,058 | 977,821 |
| C | 0 | 1M | 1M |
| C# | 0 | 1M | 1M |
| | | | |
| Total | 2,119,500 | 6,332,946 | 8,452,446 |

Table 2: Hyper-parameter settings used for pre-training

| Hyperparameter | Value |
|---|---|
| Learning Rate | 2e-4 |
| No. of epochs | 1 |
| Batch Size | 1024 |
| Warmup steps | 750 |
| Learning Rate Schedule | cosine |

## D CODE SUMMARIZATION RESULTS

In Figure 7, we show the Code summarization results on all six programming languages in the dataset.

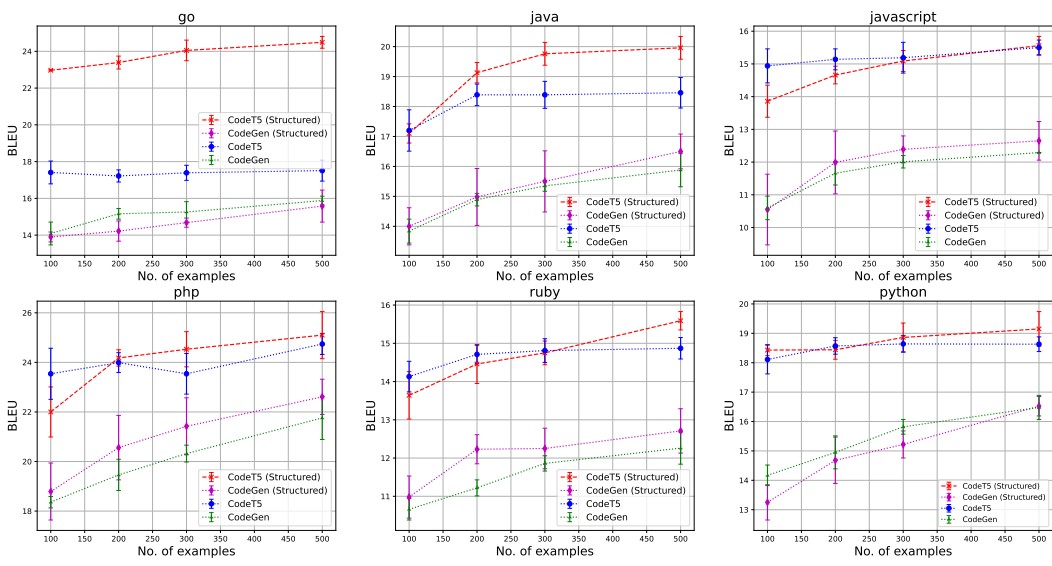

Figure 7: Code summarization BLEU scores for all six programming languages in the CodeSearch-Net benchmark.

Table 3: Hyper-parameter settings used for data-efficient fine-training

| Hyperparameter | Value |
|---|---|
| Learning Rate | 5e-5 |
| No. of epochs | 100 |
| Batch Size | 8 |
| Warmup steps | 100 |
| Learning Rate Schedule | cosine |
| Early Stopping Patience | 5 |

# E  QUALITATIVE EXAMPLES

In this section, we provide some qualitative generations from the base and structured models for certain tasks. In Listings 3, 4 and 5, we provide example test generations from all four models (CodeT5 and CodeGen, both base variants and structured models) fine-tuned with 100 training examples from the MBPP dataset. These generations are for cases where neither of the base variants produces functionally correct code, but the structured models do.

```
1  ###########################################################
2  ##############          MBPP Prompt          ##############
3  ###########################################################
4
5  Write a function to sort a list of tuples using lambda.
6
7  ###########################################################
8  ##############          CodeT5 (Base)        ##############
9  ###########################################################
10
11 def sort_list(test_list):
12   res = sorted(test_list)
13   return (res)
14
15 ###########################################################
16 ###########          CodeT5 (Structured)        ###########
17 ###########################################################
18
19 def sort_list ( test_list ) :
20    res = list ( sorted ( test_list , key = lambda tup : tup [ 1 ] ) )
21    return res
22
23 ###########################################################
24 ##############          CodeGen (Base)        ##############
25 ###########################################################
26
27 def sort_tuples(l):
28   sort_list = list(reversed(l))
29   return sort_list
30
31 ###########################################################
32 ###########          CodeGen (Structured)        ##########
33 ###########################################################
34
35 def sort_list ( l ) :
36    l . sort ( key = ( lambda x : x [ 1 ] ) )
37    return l
```

Listing 3: Models generations for a test sample in the MBPP dataset

```
1  ###########################################################
2  ##############          MBPP Prompt          ##############
3  ###########################################################
```

```
 4
 5  Write a function to remove all the tuples with length k.
 6
 7  ############################################################
 8  ###############         CodeT5 (Base)       ###############
 9  ############################################################
10
11  def remove_length(test_list, K):
12    temp = [ele for ele in test_list]
13    temp.sort()
14    res = [ele for ele in temp if len(ele) != K]
15    return (res)
16
17  ############################################################
18  ###########         CodeT5 (Structured)       ###########
19  ############################################################
20
21  def remove_length ( test_list , K ) :
22      res = [ ele for ele in test_list if len ( ele ) != K ]
23      res = list ( filter ( lambda ele : len ( ele ) != K , res ) )
24      return ( res )
25
26  ############################################################
27  ###############         CodeGen (Base)       ###############
28  ############################################################
29
30  def remove_length(test_list, K):
31    res = []
32    for sub in test_list:
33      if len(sub) != K:
34        res.append(sub)
35    res = list(set(res))
36    res = list(filter(lambda x: x != [], res))
37    res = list(filter(lambda x: x, res))
38    return (res)
39
40  ############################################################
41  ###########         CodeGen (Structured)       ###########
42  ############################################################
43
44  def remove_length_k ( nums , k ) :
45      result = list ( filter ( lambda x : len ( x ) != k , nums ) )
46      return ( result )
```

Listing 4: Models generations for a test sample in the MBPP dataset

```
 1  ############################################################
 2  ###############         MBPP Prompt       ###############
 3  ############################################################
 4
 5  Write a function to flatten a given nested list structure.
 6
 7  ############################################################
 8  ###############         CodeT5 (Base)       ###############
 9  ############################################################
10
11  def flatten_list(l):
12      result = [l]
13      for m, n, o in zip(l):
14          if (m == n == o):
15              result.append(m)
16      return result
17
18  ############################################################
19  ###########         CodeT5 (Structured)       ###########
```

```
20  ###########################################################
21
22  def flatten_list ( l ) :
23      result = [ ]
24      for ele in l :
25        if isinstance ( ele , list ) :
26          result . extend ( flatten_list ( ele ) )
27
28        else :
29          result . append ( ele )
30
31
32      return result
33
34  ###########################################################
35  ###############       CodeGen (Base)       ###############
36  ###########################################################
37
38  def flatten_list(nested_list):
39    return [item for sublist in nested_list for item in sublist]
40
41  ###########################################################
42  ############        CodeGen (Structured)       ############
43  ###########################################################
44
45  def flatten_list ( l ) :
46      res = [ ]
47      for i in l :
48        if isinstance ( i , list ) :
49          res . extend ( flatten_list ( i ) )
50
51        else :
52          res . append ( i )
53
54
55      return ( res )
```

Listing 5: Models generations for a test sample in the MBPP dataset

