# OpenReview forum: "Structured Fine-Tuning Enables Data-Efficient Adaptation of Code Language Models"
_ICLR.cc/2024/Conference — ICLR 2024 Conference Withdrawn Submission_

### Official Review · Reviewer_t7DZ · 2023-10-27

**Soundness:** 2 fair
**Presentation:** 4 excellent
**Contribution:** 2 fair
**Rating:** 5
**Confidence:** 4

**Summary:**

Most language models trained on code treat their inputs as plain-text tokens, which disregards the structured nature of programs. This work proposes to train LMs with a fairly low-level tree representation, serialized to text. Adding this formulation to both encoder-decoder and decoder-only models (in slightly different forms) increases the training speed on a range of tasks and offers benefits in some tasks after 1K training examples.

**Strengths:**

Overall, this work offers promising initial results. Compared to the (large body) of recent work on using the tree and graph representations of code, this paper chooses a fairly low-level and easily serializable tree representation. That makes the approach relatively flexible and may contribute to its positive performance. The tasks defined for CodeT5 are reasonable and the methodology was easy to follow.

The results generally paint a positive picture; focusing just on exact match (for reasons discussed below), the most substantial gains are seen by CodeT5 on a few benchmarks, while on others the performance is about even with vs. without structural information.

The paper is quite well written.

**Weaknesses:**

While the BLEU score results are very strong, the exact match rate is much less so. The structured models are often no better than their plain counterparts on the latter. This raises serious concerns around the correctness of the BLEU score implementation. One possibility is that it was measured without removing the non-terminal tokens from the serialized representations, which would make it relatively easier to predict matching tokens but not more useful for the downstream task. It is hard to tell from the implementation if this happened -- I found a flag that appears to toggle this behavior, but it is not clear if it was used. In any case, the gap is large enough to both raise concerns around the methodology and, if it is sound, create the impression that BLEU score is quite a poor proxy for performance here, since the exact match rate is virtually unchanged.

This aside, a strong downside of using tree-like representations of code is that they make the models less (or not at all) helpful in partial code contexts, where a syntax tree cannot be unambiguously parsed. The proposed format  is no exception. This rules out its usage for much of the most common use-case of LLMs for code (code completion), among others. The chosen representation also roughly doubles the length of a given program when encoded as tokens. This comes at a substantial cost given the quadratic complexity of attention involved in Transformers, and the limitations commonly imposed on context windows.

Finally, the work experiments with rather small models. While this is understandable from a resource usage perspective, it is not clear that the results will generalize to billion-scale LMs. A series of scaling results, even on sub-B parameter models, would be helpful here.

Minor notes:
- Fig. 1 does not include whitespace tokens, which might be helpful to illustrate the point made about formatting on page 4.
- P4: the choice of x, y, z to refer to important elements is somewhat surprising; I would suggest finding more distinctive tokens for these.

**Questions:**

Please consider the weaknesses above, and in particular, answer:

- Is the BLEU score implemented and evaluated only on the original code tokens? If so, what would cause there to be such a large gap in BLEU score but none in exact match? And given that, is BLEU score relevant? Perhaps provide examples where a snippet generated by a structured model has a much higher BLEU score and is also more useful, despite being inexact.
- How do you assess the cost/benefit trade-off of (a) requiring parseable code and (b) increasing the length of the program's representation in terms of tokens given that the most common use-case of LLMs for code is code generation, often in incomplete contexts.

---

### Official Review · Reviewer_28Uu · 2023-10-31

**Soundness:** 4 excellent
**Presentation:** 3 good
**Contribution:** 3 good
**Rating:** 8
**Confidence:** 4

**Summary:**

This paper addresses LLMs used as code generators. The main objective is to introduce structured knowledge in LLMs. This is done by linearizing trees representing codes and proposing many loss functions that take into consideration linearized trees.

**Strengths:**

- The paper shows that including prior structured knowledge is important
- Experiments are solid and show what is the real impact of adding structured knowledge of code

**Weaknesses:**

- The way of including structured knowledge is by using sequences of tokens that are treated as sequences of tokens. Indeed, trees are serialized
- No other ways of encoding trees are explored

**Questions:**

There is no comparison with other ways to include structures. Can you comment on this?
Moreover, structures have been largely used in neural networks for NLP, e.g., TreeLSTM https://aclanthology.org/N16-1035/, KERMIT https://aclanthology.org/2020.emnlp-main.18/. How do the current work compares with those models?



Minor: page 4, just after equation 1. k should be t

---

### Official Review · Reviewer_cViY · 2023-11-01

**Soundness:** 1 poor
**Presentation:** 2 fair
**Contribution:** 3 good
**Rating:** 3
**Confidence:** 5

**Summary:**

This paper focuses on the code language models. The authors explore a data-efficient adaptation of pre-trained code language models by further training and fine-tuning them with program structures. They represent programs as parse trees, refine a model with serialized parse trees, and fine-tune with structures. The experiments were conducted on several established datasets. The results show the effectiveness of the proposed approach.

**Strengths:**

1. The authors focus on an important area.
2. To incorporate code structure in pretraining tasks is interesting.

**Weaknesses:**

1. The experimental results may have some issues.
2. The authors missed some related work.

**Questions:**

The authors focus on an important area of code-related tasks. Existing pretraining models fail to consider the well-defined and unambiguous structures inherent in programming languages. To address this problem, the authors proposed their approach. The idea is overall interesting. However, I find some issues in this paper.

1. The experimental results may have some issues.

The main issue is in the experiments. I find that in the experiments, the CodeT5 only achieves 27.5+ BLEU, 30+ CodeBLEU, and 16+ EM in the Concode dataset. However, in their paper[1], they achieved 41.48 BLEU, 44.10 CodeBLEU, and 22.70 EM, respectively. This phenomenon also exists in other tasks in this paper. The performance in this paper even performs worse than non-pretrained structure models [7]. I doubt the reliability of the experiments, which makes the entire paper unconvincing. Further, I find the CodeT5 (Structured) even performs worse than the original CodeT5 in EM of the Concode dataset. This cannot be strong evidence to show the effectiveness of the proposed approach.

2. The authors missed some related work.

The authors aim to incorporate code structure in neural networks. However, there are some related work that is missed (e.g., [2-8]).



[1] Wang, Y., Wang, W., Joty, S., & Hoi, S. C. (2021). Codet5: Identifier-aware unified pre-trained encoder-decoder models for code understanding and generation. arXiv preprint arXiv:2109.00859.

[2] Rabinovich, M., Stern, M., & Klein, D. (2017, July). Abstract Syntax Networks for Code Generation and Semantic Parsing. In Proceedings of the 55th Annual Meeting of the Association for Computational Linguistics (Volume 1: Long Papers) (pp. 1139-1149).

[3] Yin, P., & Neubig, G. (2017, July). A Syntactic Neural Model for General-Purpose Code Generation. In Proceedings of the 55th Annual Meeting of the Association for Computational Linguistics (Volume 1: Long Papers) (pp. 440-450).

[4] Dong, L., & Lapata, M. (2016, August). Language to Logical Form with Neural Attention. In Proceedings of the 54th Annual Meeting of the Association for Computational Linguistics (Volume 1: Long Papers) (pp. 33-43).

[5] Sun, Z., Zhu, Q., Mou, L., Xiong, Y., Li, G., & Zhang, L. (2019, July). A grammar-based structural cnn decoder for code generation. In Proceedings of the AAAI conference on artificial intelligence (Vol. 33, No. 01, pp. 7055-7062).

[6] Yin, P., & Neubig, G. (2018, January). TRANX: A Transition-based Neural Abstract Syntax Parser for Semantic Parsing and Code Generation. In Proceedings of the Conference on Empirical Methods in Natural Language Processing (Demo Track).

[7] Sun, Z., Zhu, Q., Xiong, Y., Sun, Y., Mou, L., & Zhang, L. (2020, April). Treegen: A tree-based transformer architecture for code generation. In Proceedings of the AAAI Conference on Artificial Intelligence (Vol. 34, No. 05, pp. 8984-8991).

[8] Zhu, Q., Sun, Z., Zhang, W., Xiong, Y., & Zhang, L. (2022). Grape: Grammar-Preserving Rule Embedding. In IJCAI (pp. 4545-4551).

---

### Official Review · Reviewer_adVs · 2023-11-01

**Soundness:** 2 fair
**Presentation:** 3 good
**Contribution:** 2 fair
**Rating:** 5
**Confidence:** 4

**Summary:**

This paper first proposes a data-efficient adaptation method for the pre-trained code language models. Specifically, it
represents the code programs as parse trees, called CSTs, and refines a model with serialized CSTs. Then it encourages the model to learn not only the associations of code text in different languages but also the mappings of their structures and grammars, with a small number of samples. The designed framework is applicable to both the encoder-decoder and decoder-only architectures. Comprehensive comparable experiments show the effectiveness of the designed model.

**Strengths:**

1. This model framework is flexible and can be applied to both encoder-decoder and decoder-only models.
2. The paper is well-organized and well-written with clear motivations, detailed discussion, nice figures, and sufficient comparison experiments, making it easy to follow and understand.
3. This work performs comprehensive experiments over benchmark data to show the effectiveness and efficiency.

**Weaknesses:**

1. The proposed data-efficient adaption method for the pre-trained code language models is not novel. The main idea of the model design is to convert the source code to the serialized form of CSTM, which I believe is not solid and sound enough.
2. This work converts the source code into the serialized form of CST and further feeds these CST into LLMs. I am curious about the motivation for the design of CST. I know some existing methods convert source code into some formats. But what are the differences between CST and these existing methods?  What are the advantages of CST?
3. This work claims that the model is much more efficient than existing methods. I would suggest that this work discusses the complexity of these methods to prove their efficiency.

**Questions:**

1. What is the novelty of the model design? I do not think the serialized form of CSTM can be considered as the main novelty of this work.
2. What are the differences between CST and these existing methods?
3. What are the advantages of CST?
4. Discuss the complexity of these methods to prove their efficiency.

---

### Official Review · Reviewer_kKKV · 2023-11-05

**Soundness:** 2 fair
**Presentation:** 3 good
**Contribution:** 2 fair
**Rating:** 3
**Confidence:** 5

**Summary:**

This paper presents structured finetuning for coding language models. Specifically, they represent programs as concrete syntax trees (CSTs), and perform continual pretraining and finetuning with the serialized CSTs. For encoder-decoder architectures, they also present several pretraining objectives using the tree structures, including masked subtree prediction and masked node prediction. They evaluate on several benchmarks including those for code translation, code generation and code summarization, and demonstrate that finetuning with the CST format improves the performance over the text-only baselines.

**Strengths:**

1. Leveraging the code structure is a promising direction for improving coding language models.

2. The authors conduct experiments on multiple important domains for coding applications.

**Weaknesses:**

1. The novelty of the approach is very limited, with several missing references. First, the idea of representing programs as trees is not new. In particular, the paper misses the discussion of [1], which proposes to encode the input code in its syntax tree format, and also decode the syntax tree for code translation. On the other hand, [2] and other works cited in the paper (e.g., UniXCoder) also designed pretraining objectives that utilize the tree structure.

2. This work argues that using CST is better than other formats such as AST; however, there is no empirical evidence to justify this claim. Adding this comparison is helpful, especially since AST has been used by several prior works.

3. For code translation, it is better to use those benchmarks with unit tests, such as Transcoder [3] and MBXP [4], as other metrics without execution are less reliable.

4. For MBPP results, the CodeGen paper reports a pass@1 of 7.46, which is much higher than the baseline reported in Figure 3. What causes the discrepancy? Also, it is helpful to report results with more samples, e.g., pass@10 and pass@100 as in the CodeGen paper.

[1] Xinyun Chen, Chang Liu, Dawn Song, "Tree-to-tree Neural Networks for Program Translation", Advances in Neural Information Processing Systems, 2018.
[2] Xin Wang, Yasheng Wang, Fei Mi, Pingyi Zhou, Yao Wan, Xiao Liu, Li Li, Hao Wu, Jin Liu, Xin Jiang, "SynCoBERT: Syntax-Guided Multi-Modal Contrastive Pre-Training for Code Representation".
[3] Marie-Anne Lachaux, Baptiste Roziere, Lowik Chanussot, Guillaume Lample, "Unsupervised Translation of Programming Languages
", NeurIPS 2020.
[4] Ben Athiwaratkun et al., Multi-lingual Evaluation of Code Generation Models, ICLR 2023.

**Questions:**

1. Please clarify the novelty of this work and add missing references [1] [2].

2. Please provide empirical comparison to justify the claim that using CST is better than other formats such as AST.

3. For code translation, please use those benchmarks with unit tests, such as Transcoder [3] and MBXP [4].

4. Please clarify the result discrepancy on MBPP, and report results with more samples, e.g., pass@10 and pass@100.

[1] Xinyun Chen, Chang Liu, Dawn Song, "Tree-to-tree Neural Networks for Program Translation", Advances in Neural Information Processing Systems, 2018.
[2] Xin Wang, Yasheng Wang, Fei Mi, Pingyi Zhou, Yao Wan, Xiao Liu, Li Li, Hao Wu, Jin Liu, Xin Jiang, "SynCoBERT: Syntax-Guided Multi-Modal Contrastive Pre-Training for Code Representation".
[3] Marie-Anne Lachaux, Baptiste Roziere, Lowik Chanussot, Guillaume Lample, "Unsupervised Translation of Programming Languages
", NeurIPS 2020.
[4] Ben Athiwaratkun et al., Multi-lingual Evaluation of Code Generation Models, ICLR 2023.

---

### Author Response · Authors · 2023-11-22

We thank the reviewers for taking the time to review our work. We received useful critical feedback that we believe with help us improve our work.

Multiple reviewers raised concerns about mismatch between results reported in our work vs. results reported in previous works, and we would like to address these concerns:
- **Reviewer kKKV asked about the discrepancy in the MBPP benchmark results**
	- The prompt used in original CodeGen evaluation for this task is of the following format:
	```
	def similar_elements(test_tup1, test_tup2):
        '''
        Write a function to find the shared elements from the given two lists.
        '''
	```
	- This prompt formulation requires the model to complete the function body given the function signature and docstring.
  - We, on the other hand, formulated the problem as a stricter text-to-code task, where the input is the docstring alone and the output is the entire Python function.
  - Due to this difference in task formulation, we observe the discrepancy in the results.
- **Reviewer cViY raised concern about mismatch in CodeT5 results**
  - The reviewer pointed out that for the Concode benchmark, we report BLEU=27.5+, CodeBLEU=30+ and EM=16+, while the original CodeT5 paper reports BLEU=41.48, CodeBLEU=44.10, and EM=22.70. The reviewer also pointed out that this phenomenon exists in other tasks, and thus questioned the reliability of the experiments.
  - We respectfully point-out that this is due to a misinterpretation of the two results.
  - Results reported in our work are for varying number of fine-tuning examples (100-1000 fine-tuning examples), while the results reported in prior works are when the entire training dataset of the benchmark is used.
  - Specifically for the Concode dataset, the results reviewer quoted from our work (BLEU=27.5+, CodeBLEU=30+ and EM=16+) are when only 1000 fine-tuning examples are used, and the results reported in the CodeT5 paper (BLEU=41.48, CodeBLEU=44.10, and EM=22.70) are when the entire 100,000 fine-tuning examples in the Concode dataset are used. Because we use 1% of the original training dataset, the results are bound to be different. This is the case with other results reported in the paper as well.
  - We would also like to point out that while we use a subset of the training data to fine-tune models, we use the entire test dataset for each benchmark to evaluate and report the results.
- **Reviewer t7DZ asked about the correctness of BLEU score implementation**
  - We confirm that the BLEU score implementation is correct and is computed on the code (only non-terminal nodes) and not on the serialized trees (both terminal and non-terminal nodes). Along with BLEU scores, we also report CodeBLEU and pass@k metrics which can only be computed on code and not on trees. We show improvements similar to BLEU scores on these metrics as well.
  - We acknowledge that BLEU score is not an appropriate metric to evaluate code and it has been highlighted in various prior works as well. However, we chose to report the metric to ensure consistency between metrics reported for these benchmarks in prior works and our work.


Having clarified the concerns, we appreciate feedback provided by the reviewers. Specifically,
- Reviewer kKKV suggestion to include benchmarks with unit-tests and MBPP results with more samples
- Reviewer adVs, kKKV, 28Uu, cViY suggestion to compare with other ways of including structures and related work
- Reviewer t7DZ suggestions to include scaling experiments

We believe these experiments will help improve our work, but we require more time to include these experiments appropriately in the paper. Thus, we have decided to withdraw the submission.